# Breast Reconstruction by Exclusive Lipofilling after Total Mastectomy for Breast Cancer: Description of the Technique and Evaluation of Quality of Life

**DOI:** 10.3390/jpm12020153

**Published:** 2022-01-25

**Authors:** Alexandre Piffer, Gabrielle Aubry, Claudio Cannistra, Nathalie Popescu, Maryam Nikpayam, Martin Koskas, Catherine Uzan, Jean-Christophe Bichet, Geoffroy Canlorbe

**Affiliations:** 1Department of Gynecological and Breast Surgery and Oncology, Assistance Publique des Hôpitaux de Paris (AP-HP), Pitié-Salpêtrière University Hospital, 75013 Paris, France; alexandre.piffer@aphp.fr (A.P.); gabrielle.aubry@aphp.fr (G.A.); nathalie.popescu@aphp.fr (N.P.); marianne.nikayam@aphp.fr (M.N.); catherine.uzan@aphp.fr (C.U.); jeanchristophe.bichet@aphp.fr (J.-C.B.); 2Department of Gynecology, Assistance Publique des Hôpitaux de Paris (AP-HP), Hôpital Bichat, 75018 Paris, France; claudio.cannistra@aphp.fr (C.C.); martin.koskas@aphp.fr (M.K.); 3Centre de Recherche Saint-Antoine (CRSA), INSERM UMR_S_938, Cancer Biology and Therapeutics, Sorbonne University, 75012 Paris, France

**Keywords:** breast cancer, breast reconstruction, lipofilling, fat grafting, reconstructive surgery

## Abstract

Background: The objective of this work was to describe the technique of exclusive lipofilling in breast reconstruction after total mastectomy, to evaluate the satisfaction and quality of life of the patients, and to explore current literature on the subject. Methods: We conducted a retrospective observational multicentric study from January 2013 to April 2020. The modalities of surgery, esthetic result, and patient satisfaction were evaluated with the breast reconstruction module of BREAST-Q. Results: Complete data were available for 37 patients. The mean number of sessions was 2.2 (standard deviation 1.1), spread over an average of 6.8 months (SD 6.9). The average total volume of fat transferred was 566.4 mL. The complication rate was 18.9%. No severe complication was observed (Clavien–Dindo 3/4). Two patients were diagnosed with recurrence, in a metastatic mode (5.4%). The average satisfaction rate was 68.4% (SD 24.8) for psychosocial well-being and 64.5% (SD 24.1) for sexual well-being. The satisfaction rate was 60.2% (SD 20.9) for the image of the reconstructed breast and 82.7% (SD 21.9) for locoregional comfort. Conclusions: Breast reconstruction by exclusive lipofilling after total mastectomy provides satisfactory quality of life scores. The simplicity of the surgical technique and equipment required, and the high satisfaction rate confirm that lipofilling should be included in the panel of choice of breast reconstruction techniques.

## 1. Introduction

With nearly 2,261,000 new cases and causing 685,000 deaths worldwide in 2020, breast cancer is the most common cancer in women and therefore a real public health issue [1]. Partial or total mastectomy can alter a woman’s body shape and have a socio-professional, sexual, and psychological impact [2,3,4,5,6,7,8,9,10,11]. A better quality of life has been reported in patients who have had breast reconstruction after total mastectomy [12,13,14]. However, the number of total mastectomy patients initiating breast reconstruction remains relatively low: the overall reconstruction rate is estimated at 56% in the United States and 35% in France [15,16]. This low rate of reconstruction can be explained mainly by the age of the patient, the place of care, difficulty of access to plastic surgery techniques, or the apprehension of a heavy surgical technique involving the implantation of a foreign body [15,16]. Since the 2000s, a new reconstructive surgery technique—called autologous fat grafting, or lipofilling—has been increasingly used throughout the world and has completely changed the approach to breast reconstruction [17,18,19]. The technique is easy to perform and is very well tolerated. However, until recently, lipofilling has mainly been used as an add-on to perfect the results of other methods of breast reconstruction: for example, to cover an implant, or correct volume or asymmetry defects. 

This is now changing, and there is a boom in exclusive lipofilling breast reconstruction with positive empirical results reported by patients. Nevertheless, data in the literature on exclusive lipofilling in breast reconstruction are almost nonexistent, and the few studies available concern small numbers of patients and do not evaluate quality of life. The objective of this work was to describe the technique of exclusive lipofilling in breast reconstruction after total mastectomy, to evaluate the satisfaction and quality of life of the patients, and to explore current literature on the subject.

## 2. Materials and Methods

### 2.1. Study Design

We conducted a retrospective observational study in the Gynecological and Breast Surgery and Oncology Department of Pitié-Salpêtrière Hospital (APHP), the private Rémusat Clinic in Paris, and in the Department of Gynecology of Bichat Hospital (APHP) from January 2013 to April 2020. The patients included were all informed of the study, and all agreed to participate. The study protocol was accepted by the French Committee on Ethics and Research in Obstetrics and Gynecology (CEROG 2021-GYN-0410).

The inclusion criteria were as follows: all adult patients who had undergone exclusive lipofilling breast reconstruction after total mastectomy. The exclusion criteria were patients who had undergone lipofilling in addition to, or in preparation for, another breast reconstruction technique (breast implant or autologous flap), and patients whose breast reconstruction was not completed. Epidemiological and clinical data were collected from the computerized records from routine patient care: age at the time of the first lipofilling surgery, body mass index (BMI), bra size and cup, smoking, medical history, breast cancer characteristics (tumor size and histological profile, tumor-node-metastasis (TNM) classification), adjuvant and neoadjuvant treatments received (chemotherapy, hormonal therapy, radiation therapy). Data concerning surgery were also collected and evaluated: the time between the end of the oncologic treatment (surgery or radiotherapy) and the start of the lipofilling procedure, scheduling of the sessions and number of sessions, the volume of fat removed and reinjected per session, nipple–areolar reconstruction, and any corrective measures undertaken to ensure symmetry with the contralateral breast.

Quality of life data were obtained from the validated French version of the breast reconstruction module of BREAST-Q, which all the patients completed during postoperative consultations. The postoperative breast reconstruction module of the BREAST-Q [20,21] assesses a patient’s physical and psychosocial quality of life and sexual well-being, as well as satisfaction with the care in patients who have had breast surgery. The questionnaire provides a quality of life score separately for each module on a scale of 0 to 100. The higher the score, the greater the satisfaction.

### 2.2. Surgical Technique

The surgeries were all performed by experienced hospital practitioners specializing in oncologic and reconstructive breast surgery. In our department, all reconstruction techniques (prosthesis, flap, lipofilling) are systematically proposed to our patients. The advantages and disadvantages of each technique were explained to the patients and final decision was taken after discussion. The final choice of technique was made after discussing with the patient and according to her wishes and morphology. The modality of reconstruction was systematically adapted to the morphology of each patient and her comorbidities. Surgery was performed on an outpatient basis if all conventional criteria for outpatient surgical management were met. Donor sites for the first lipofilling session were determined at the time of the preoperative consultation based on the patient’s morphology, and fat was harvested from those sites thereafter. The patients were positioned in the dorsal decubitus position, arms in the form of a cross, and under general anesthesia. The procedure started with infiltration of the donor site with adrenaline saline (1 mg of adrenaline for 1 L of saline). The technical set-up consisted of two conventional suction tubes to connect the sampling cannula to a 200 mL Redon bottle, itself connected to a wall suction. The fat collected by liposuction directly in the Redon was then distributed in syringes of 10 mL which were centrifuged for 1 min at 900 G (3000 rpm). The purified fat obtained was separated from the liquid phase, which was not preserved. The patient was installed in semi-seated position for reinjections: the purified fat was reinjected by means of a 2.5 mm metal cannula with a blunt tip, in the entire breast projection area in a retro-traceable and radial manner in the different planes (pectoral and subcutaneous), through several incisions previously made with a needle. For patients benefiting from immediate reconstructions, the fat was reinjected directly into the pectoral muscle. During the additional sessions, the fat was then reinjected into the pectoral muscle and the subcutaneous layer, as in the case of secondary breast reconstruction patients. For secondary breast reconstruction, the skin was freed from the chest wall by multiple fasciotomies alternating with the reinjection of the purified fat. Once the fat was reinjected, a fat-containing dressing was applied. Depending on the esthetic result, new sessions could be offered at 3-month intervals. The patients were then seen in consultation at 15 days, 3 months, and then every 6 months as part of the breast cancer follow-up.

### 2.3. Statistical Analysis

Categorical variables are described as a percentage of the total. Quantitative variables are analyzed by their mean and standard deviation (SD). The statistical analysis was performed on Excel software 

### 2.4. Literature Research

A literature search was performed using the PubMed database. The following search terms were used: (mastectomy) AND (fat graft * OR lipofilling OR fat transfer OR lipotransfer OR fat transplantation OR lipomodelling). Studies found manually or through the reference lists of included studies were also eligible for inclusion. Studies were included if the patients underwent breast reconstruction with fat grafting as the only treatment modality after the surgical resection of the whole breast due to cancer or a genetic predisposition to breast cancer (e.g., BRCA mutation).

The following baseline characteristics were collected from each study: author, publication year, study design, number of patients. Data specific to the lipofilling procedure were also collected and included the following: the number of fat grafting treatments needed to complete a reconstruction, accumulated fat graft volume, fat graft volume per procedure, percentage and type of complications, expansion method of the recipient site, and available data concerning satisfaction.

## 3. Results

### 3.1. General Results

During the study period, 50 patients underwent breast reconstruction by lipofilling in the three centers. Of these, 37 (71%) had completed their breast reconstruction in two centers (Pitié-Salpêtrière and Rémusat Clinic) and were included in the data analysis. The other 15 patients (29%) were still undergoing reconstruction at the end of the inclusion period and were excluded from the analysis. Demographic data and tumor characteristics are shown in Table 1 and Table 2. The mean age of the patients included was 48 years (SD 10.2), with a mean BMI of 23.9 kg/m^2^ (SD 5.2, (38–75 years)). Ten patients (27%) were active smokers. The most frequent comorbidities were high blood pressure (10.1%) and type 2 diabetes (5.4%). Most of the patients (29/37, 78.3%) had an A or B cup.

### 3.2. Oncological Aspects

The indication for mastectomy was invasive ductal carcinoma for twenty-six patients (62.1%) and ductal carcinoma in situ for six (16.2%). Twenty-one patients (56.7%) received neoadjuvant and/or adjuvant chemotherapy. Nineteen patients (51.3%) received additional radiation and twenty-one (56.7%) received hormone therapy.

An axillary lymph node procedure was performed in thirty-six patients (98%). The only patient who did not undergo axillary exploration had a phyllodes tumor.

### 3.3. Technical Aspects 

Breast reconstruction was immediate in four (10.8%) patients and secondary in thirty-three (89.2%). A contralateral symmetrization procedure, such as the repair of breast ptosis with or without reduction, was performed in seventeen patients (45.9%). Among the patients with delayed reconstruction, the mean time from completion of treatment (total mastectomy or radiotherapy) to the first session of lipofilling reconstruction was 14.8 months (SD 8.9). The mean number of sessions was 2.2 (SD 1.1) spread over an average of 6.8 months (SD 6.9). Among the 10% of patients who had more than three operations were those of C or D cup, which required a bigger fat transfer. Overall, the mean total volume transferred was 566.4 mL (SD 441.6). The mean retrieval and reinjection volumes per session were 520.2 mL (SD 81.7) and 257.4 mL (SD 46.7), respectively. Nipple reconstruction was performed or scheduled for twelve patients (32.4%). The procedure took place in an outpatient setting for 30 patients (80%). For these patients, there was no conversion to conventional hospitalization, nor any hospitalization in the days following the surgery. There were no peroperative complications. Seven patients (18.9%) experienced a postoperative complication, none of which were severe (Clavien–Dindo grade 1 or 2) [22]. Four patients had an oil cyst which resolved on puncture; two had cytosteatonecrosis which did not require specific management; finally, one patient was consulted for a fistula at one of the reinjection points, which was managed by a single stitch with a resorbable rapid polyglactin 5-0 suture. Patients’ smoking did not present an increased complication rate compared to nonsmoking patients. These results are summarized in Table 3.

### 3.4. Patient Quality of Life and Satisfaction

Twenty-five patients (67%) completed the breast reconstruction module of the BREAST-Q. The questionnaire was completed only once during a postoperative consultation after the end of the reconstruction. The average time from the end of the reconstruction to the completion of the questionnaire was 22.1 months (SD 15.3). The average satisfaction rate for psychosocial well-being was 68.4% (SD 24.8) and 64.5% (SD 24.1) for sexual well-being. The average rate of satisfaction for the image of the reconstructed breast was 60.2% (SD 20.9) and 82.7% (SD 21.9) for locoregional comfort related to the technique itself. For the patients who had a nipple reconstruction, 80% of patients were satisfied or very satisfied. The average scores for the quality of information provided by the surgeon, the paramedical team, and the administrative staff were, respectively, 80%, 80%, and 83% (SD 33, 31, 24). Figure 1 shows the results after total mastectomy and after two sessions of secondary breast reconstruction by exclusive lipofilling. Figure 2 and Figure 3 show the results after total mastectomy and after three sessions of secondary breast reconstruction by exclusive lipofilling. The scar in the sub mammary fold in Figure 3 was a small pexia to reshape the breast and make it more symmetrical to the other breast, which had been lifted with an inverted-T reduction surgery.

### 3.5. Survival Outcome 

The mean follow-up time at the end of the oncological treatment, surgery or radiotherapy, and latest news was 43.7 months (SD 19.4). Tumor recurrence was diagnosed in two patients (5.4%) at 13 and 21 months after the end of radiotherapy and consisted of distant metastases for both (hepatic and contralateral supraclavicular lymph nodes). Both cases were infiltrating ductal carcinoma with initial axillary lymph node involvement. The first case was an HER overexpressing cancer and the second was a triple negative cancer. None of them had a BRCA mutation. 

### 3.6. Literature Review

The initial search identified 545 studies. After screening based on title and abstract, 37 studies were eligible for full text screening. The full text screening excluded 22 studies. The screening process is shown in Figure 4.

The study characteristics, mostly case reports and retrospective studies, are shown in Table 4.

## 4. Discussion

Breast reconstruction by exclusive lipofilling after total mastectomy provides satisfactory quality of life scores at 22 months after the completion of the reconstruction. Furthermore, our results confirm that the technique is simple and associated with an absence of peroperative complications and mild postoperative morbidity.

As lipofilling is usually used as an additional tool to prepare or to improve a breast reconstruction, rather than as an exclusive reconstruction method per se, literature on the subject is scarce [24,35,43,44,45]. Nevertheless, it is an attractive method for breast reconstruction because it allows the creation of a breast with a natural consistency, which evolves with skin aging and weight variations. A significant correlation between change of weight and fat transplant volume survival over the years has been reported in the literature [46,47]. Another potential esthetic advantage may be liposuction from areas that the patients may complain about. Scarring is minimal and, in the case of failure, does not create an obstacle to attempt another reconstruction method [24,44,48]. Besides, it can be a valuable option for patients who wish to undergo breast reconstruction but who suffer from comorbidities such as diabetes or arterial disease, or who are active smokers, all of which may be contraindications for other reconstruction techniques [35,49,50,51,52,53]. The main limiting factor of this surgical technique is that the sessions are spread out over time, and the repetition of the surgical procedure which may discourage both the surgeon and the patient. In our study, the average number of sessions was 2.2 (SD 1.1) spread over an average of 6.8 months (SD 6.9). In the literature, the number of sessions ranges from 2 to 5.8 spread over a period of about 15 months [24,27,35,48]. This type of breast reconstruction could therefore be offered as a priority to patients with an A or B cup, which could limit the number of procedures and the time required for reconstruction. Some teams report the use of a skin expansion with expander or expansion device to enhance the breast (BRAVA system) [24,25,28,30,31,37,38]. In the technique we are presenting, there is no need for such systems: the skin, thanks to its plasticity, is progressively and smoothly expanded by the fat to the desired volume. Finally, breast lipofilling increases the occurrence of cytosteatonecrosis, which can make ultrasound monitoring difficult for nonspecialized teams [54,55]. In our study, it was not possible to correlate the outcome with postoperative imaging data. New imaging techniques, such as numerized or 3D mammography, or specific MRI sequences can help for understanding such images. 

The quality of life results of our work are similar to those found in other published studies concerning breast reconstruction by lipofilling. Our study found a mean score of 68.4 for psychological well-being. This is comparable to the findings of Bayti et al., who reported a score of 68 for this item [24] in their retrospective observational study of 58 patients, 22 of whom underwent breast reconstruction by exclusive lipofilling from 2009 to 2014. The team of Santosa et al. compared the satisfaction of patients after breast reconstruction by breast implant or autologous flap using the BREAST-Q. At 1 year, they found a satisfaction score of 71.8 and a psychosocial well-being score of 74.7 [56]. In terms of satisfaction with the reconstructed breast, our study found a score of 60.2% (SD 20.9), which is also comparable to the scores found by Santosa et al.: 63.1% for breast implants and 68.6% for autologous flaps [56]. In the Bayti et al. study, patients who had exclusive lipofilling reconstruction gave an average score of 52.2% for sexual quality of life, which is slightly lower than the score of 64.5% in our study. 

This score remains relatively low in comparison with the other scores on the scale, whereas quality of life studies after total mastectomy often report that sexual quality of life is negatively impacted, particularly for patients under 50 years of age, whether they have undergone reconstruction or not [5]. In Santosa et al.’s study, the sexual well-being score was 52.7% and 55.5%, respectively, for breast reconstruction by implant or autologous flap [56]. These lowish scores are probably due in part to the fact that any type of breast reconstruction does not restore the tactile sensation of the skin nor preserve the erectile function of the nipple, which are important elements for an erogenous breast. The results in terms of quality of life are therefore superposable, whatever the type of reconstruction. Finally, the scores concerning the information received, the surgeon, and the medical and paramedical team were high. Nearly 70 percent of the patients rated their satisfaction with the surgeon as the highest. Although data concerning preoperative quality of life were not at our disposal, these results show that the perioperative experience after exclusive lipofilling is very satisfactory for the patient.

All the patients in our series were operated on by the same plastic surgeon, with several years of experience in this technique as well as in other methods of reconstruction. More complications may be observed after procedures performed by lesser-experienced surgeons who have not fully mastered the technique. The complication rate was 18.9% in our series, which is similar to that observed in the literature concerning breast reconstruction by lipofilling. We had no severe complications (i.e., Clavien–Dindo ≥ 3). The complication rate after exclusive lipofilling is also lower when compared to reconstructions with implants or flaps: 15.6% versus 41.3% and 30.9%, respectively [57,58]. The complications observed are mainly the occurrence of cytosteatonecrosis, oil cysts, hematoma at the donor site, infection, or skin necrosis. These complications are usually benign and simple to manage [27] compared to complications that may occur with other techniques (implant infections, flap necrosis). Kellou et al. performed a retrospective study of 45 patients who underwent exclusive lipofilling reconstruction after total mastectomy for breast cancer. The authors compared 22 patients who had completed reconstruction with 16 patients who changed their reconstruction technique. They estimated the failure rate of exclusive lipofilling at 32.6%. In 11 cases, the lipofilling was stopped either because of excessive absorption of the reinjected fat or by insufficient fat reserve. In five cases, patients decided on their own to give up on exclusive lipofilling reconstruction. The main factors of failure were the use of adjuvant chemotherapy and an older age. In this study, radiation therapy did not influence the success rate. The main postoperative complications were the occurrence of superficial burning at reinjection site and hematoma, which we did not observe in our study [24,35].

Among the patients with delayed breast reconstruction, the mean time from completion of treatment (total mastectomy or radiotherapy) to the first session of lipofilling reconstruction was 14.8 months (SD 8.9). This is in line with the latest French recommendations, which recommend an interval of 2 years after the completion of local treatments only for women with a significant risk of local recurrence [43]. Tumor recurrence occurred in only two patients in our series, and both consisted of distant metastases. The studies available in the literature are generally reassuring regarding locoregional recurrence after lipofilling, but the follow-up in our study is too short for hormone-sensitive tumors [35,44,59,60,61,62]. It is therefore advisable to remain cautious, especially in patients with luminal A cancer: Sorrentino et al. showed an increased tendency for locoregional and distant recurrence in patients with luminal A cancer who had undergone lipofilling reconstruction after a follow-up of only 6 years. However, the authors concluded that lipofilling could be safely proposed in the setting of multimodal adjuvant treatment. [63]. Given the importance of the endocrine role of adipocytes, caution should be exercised regarding the indications for lipofilling in hormone-dependent cancers with poor prognosis criteria such as lymph node involvement [63].

Some limitations of the present study deserve to be underlined. First, the retrospective nature of the study cannot exclude any bias. Second, our series was relatively small. However, to our knowledge, this is the largest series published on the subject to date. Third, the presence of a control group would have been interesting, particularly to compare the quality of life scores. Given the satisfactory results on quality of life, an interesting parameter to evaluate would have been the medico-economic aspect of the technique. In our study, 80% of the procedures were performed as outpatient surgery, which improves comfort and satisfaction, and reduces unnecessary hospitalizations and work stoppage duration. This results in reducing health care costs (38–42). At last, unlike microsurgical flap techniques which require long occupancy of operating rooms, and unlike the prosthetic material required for reconstruction with implants, this technique is fast and does not require specific equipment. In the context of a public health issue, or in countries that do not cover the cost of breast reconstruction, these arguments support including lipofilling reconstruction in the panel of reconstruction method options. Finally, we did not study the learning curve of the technique since it was mastered for several years by the surgeons operating in our study. 

## 5. Conclusions

Breast reconstruction by exclusive lipofilling after total mastectomy provides satisfactory quality of life scores. The simplicity of the surgical technique and of the equipment required, the low complication rate, the oncological safety, and the high satisfaction rate confirm that exclusive lipofilling should be included in the panel of choice of breast reconstruction techniques. However, prospective comparative studies with other techniques are required to confirm these results.

## Figures and Tables

**Figure 1 jpm-12-00153-f001:**
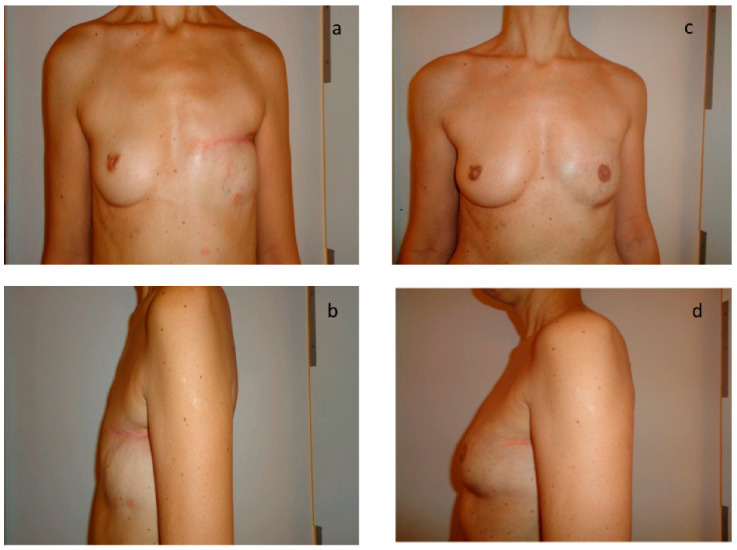
Aesthetic result after breast reconstruction by exclusive lipofilling in patient with A bra size. (**a**,**b**) Results after total mastectomy; (**c**,**d**) Results after 3 sessions of reconstruction by exclusive lipofilling.

**Figure 2 jpm-12-00153-f002:**
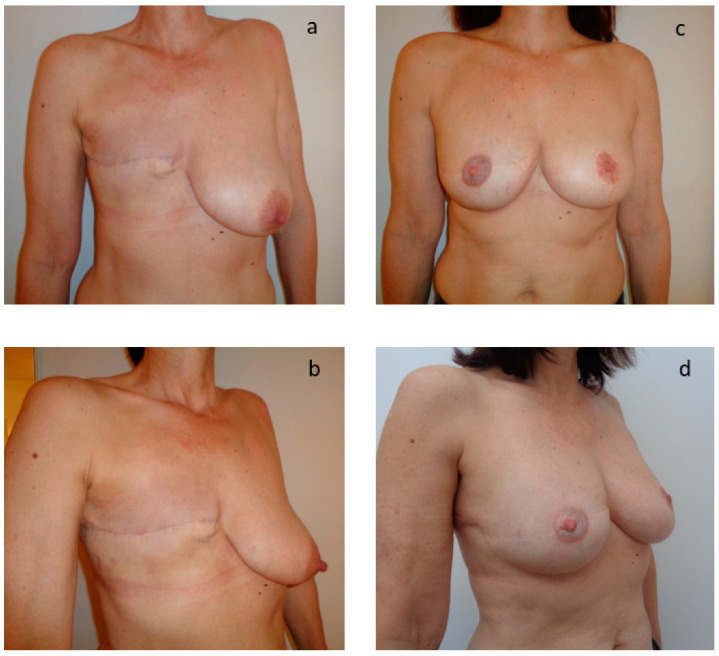
Aesthetic result after breast reconstruction by exclusive lipofilling in patient with B bra size. (**a**,**b**) Results after total mastectomy; (**c**,**d**) Results after 3 sessions of reconstruction by exclusive lipofilling.

**Figure 3 jpm-12-00153-f003:**
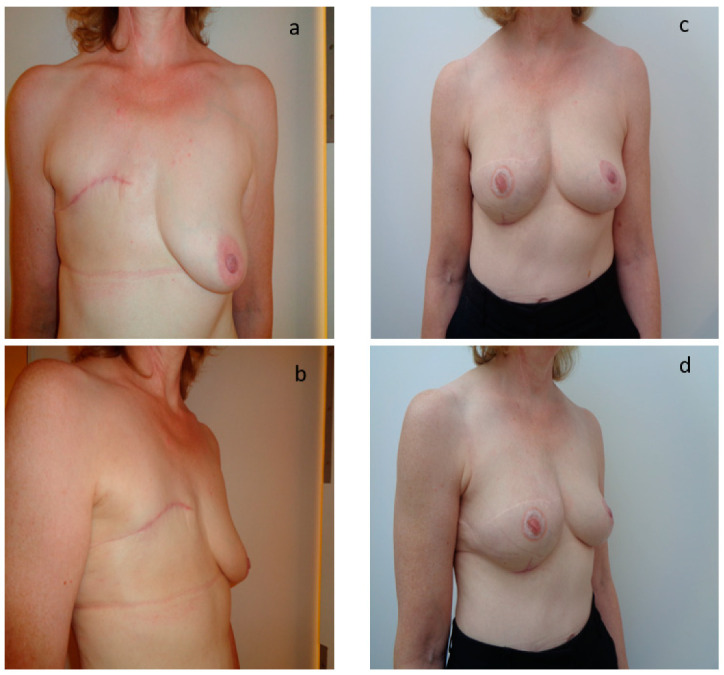
Aesthetic result after breast reconstruction by exclusive lipofilling in patient with C bra size. (**a**,**c**) Results after total mastectomy; (**b**,**d**) Results after 3 sessions of reconstruction by exclusive lipofilling. The scar in the sub mammary fold was a small pexia to reshape the breast and make it more symmetrical to the other breast, which had been lifted with an inverted-T reduction surgery.

**Figure 4 jpm-12-00153-f004:**
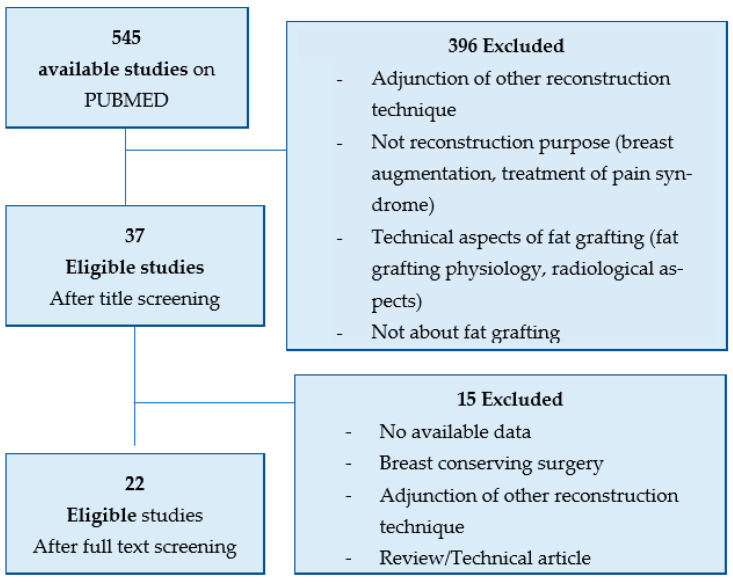
Flow chart of the screening process.

**Table 1 jpm-12-00153-t001:** General characteristics of the study population (*n* = 37).

	N (%)
Age (years)	48.3 (10.2)
BMI (kg/m²)	23.9 (5.2)
Active smoking	10 (27%)
Type 2 diabetes	2 (5.4%)
Depression	2 (5.4%)
High blood pressure	4 (10.1%)
Chest size (cm)	75.2 (7.4)
Bra cup size	
A	7 (18.9%)
B	22 (59.4%)
C	6 (16.2%)
D	1 (2.7%)
F	1 (2.7%)

Continuous data are presented as average (standard deviation). Categorial data are presented as absolute numbers (percentage). BMI: body mass index.

**Table 2 jpm-12-00153-t002:** Oncological characteristics (*n* = 37).

	N (%)
Histological types	
Ductal carcinoma invasive (DCI)	23 (62.1%)
Ductal carcinoma in situ (DCIS)	6 (16.2%)
Lobular carcinoma invasive (LCI)	5 (13.5%)
Combined breast cancer *	2 (5.4%)
Other	1 (2.7%)
Hormonal receptors	25 (67.5%)
HER 2+	23 (62%)
Tumor size according to TNM classification	
Tis	4 (10.8%)
T1	17 (45.9%)
T2	11 (29.7%)
T3	3 (8.1%)
T4	2 (5.4%)
Axillary lymph node disease	16 (43.2%)
Visceral metastases	3 (8.1%)
Chemotherapy	21 (56.7%)
Radiotherapy	19 (51.3%)
Hormotherapy	21 (56.7%)
Axillary node management	
No	1 (2.7%)
Sentinel node	15 (40.5%)
Axillary Lymph node Dissection	21 (56.7%)

Continuous data are presented as average (standard deviation). Categorial data are presented as absolute numbers (percentage). * Combined breast cancer: DCI + LCI; Tis: ductal carcinoma in situ. TNM: EPU Breast Diseases 2018.

**Table 3 jpm-12-00153-t003:** Technical results of lipofilling procedure (*n* = 37).

	N (37)
Time of breast reconstruction	
Immediate	4 (10.8%)
Delayed	33 (89.2%)
Symmetrisation procedure	17 (45.9%)
Fat volume collected per patient	
Total (mL)	1144.6 (726.8)
Per session (mL)	520.2 (81.7)
Fat volume transferred per patient	
Total (mL)	566.4 (441.6)
Per procedure (mL)	257.4 (46.7)
Number of procedures	2.2 (1.1)
1	11 (29.7%)
2	10 (27.0%)
3	12 (32.4%)
4	3 (8.1%)
6	1 (2.7%)
Staggered reconstruction (months)	6.8 (6.9)
Interval between surgery/radiotherapy–lipofilling (months) *	14.8 (8.9)
Nipple reconstruction **	12 (32.4%)
Number of complications	7 (18.9%)

Continuous data are presented as average (standard deviation). Categorial data are presented as absolute numbers (percentage). * Delayed breast reconstruction. ** Planned or already done.

**Table 4 jpm-12-00153-t004:** Characteristics of the included studies.

Author	Year	Design	Number of Patients	Number of Sessions	Total volume Injected (mL)	Volume Per Session (mL)	Expansion System	Complications	Health Quality of Life
Babin [4]	2017	Case report	1	3	500	166	None	Infection	“Satisfied”
Babovic [23]	2010	Case report	1	6	1615	403.9	None	0	“Satisfied”
Bayti [24]	2015	Retrospective	22	4.9	1421	NR	BRAVA	18.2% cytosteatonecrosis	Breast Q: 98% “very satisfied”
								4.5% hematoma	
								13.6% donor site irregularity	
Cheng [25]	2013	Case report	1	2	430	215	BRAVA	0	-
Costantini [26]	2013	Prospective	2	2	302	100.48	None	-	-
Delaporte [27]	2009	Retrospective	15	3	600	-	None	20% cytosteatonecrosis	66.7% “very good”
Fabiocchi [28]	2017	Retrospective	57	3.6	640	318	Expander	0.75% hemorrhage donor site	64.8% “excellent”
								0.75% surgical site infection	
Fitoussi [29]	2009	Case report	1	2	380	-	None	-	-
Hammer-Hansen [30]	2015	Case report	1	7	957	136.71	BRAVA	Rash	-
Ho Quoc [31]	2016	Retrospective	6		790	-	BRAVA	-	82% “satisfied”
Ho Quoc [32]	2019	Retrospective	2	4	474	118	None	-	-
Hoppe [33]	2013	Retrospective	28	5	1020	159	None	2.59% cytosteatonecrosis	96% “high
								0.74% infection	satifaction”
								0.74% granuloma	
Jarrah [34]	2013	Case report	1	3	720	240	None	-	-
Kellou [35]	2019	Retrospective	22	5.86	1490.6	-	None	50% hematoma donor site	
								22.7% lipolysis,	-
								44.4% puncture site burn,	
								11% lymphocele (donor site)	
								11% hip phlyctene	
Longo [36]	2014	Prospective	21	4	439	137	None	-	-
Manconi [37]	2017	Retrospective	12	3	417	214	Expander	3.2	-
Mestak [38]	2013	Case report	1	3	815	271	BRAVA	0	-
Niddam [39]	2017	Retrospective	25	-	-	-	None	-	Satisfaction rate: 5.8/10
Pannettiere [25]	2011	Case report	1	9	700	78	None	0	-
Serra-Renom [40]	2011	Case series	8	3	400	133	None	0	-
Stillaert [41]	2016	Case series	8	4	644	160	Expander	1 cyst	-
Zhang [42]	2020	Retrospective	30	3,3	-	230.5	+/− BRAVA	3% cellulitis 27% cysts 3% palpable nodules	90% “very satisfied”

Hyphens indicate data not available.

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
