# Peer review of "Breast Reconstruction by Exclusive Lipofilling after Total Mastectomy for Breast Cancer: Description of the Technique and Evaluation of Quality of Life"

_jpm, 2022, doi:10.3390/jpm12020153_

Round 1

Reviewer 1 Report

The technique is accurately described and the literature review is comprehensive.
Feedback from patients regarding their personal satisfaction with the treatment was evaluated by the Breast-Q in a fair manner.

The only thing that needs further investigation is the volume of fat inserted with lipofilling and the residual volume at 1 year.

Author Response

We thank the reviewer for his insightful suggestion.

Actually, the volume of fat inserted with lipofilling is described in Table 3: “Fat volume transferred per patient” with 2 lines: “Total” and “Per procedure”.

Concerning the residual volume at 1 year, the volume of the reconstructed breast was stable, no patients complained of any loss of volume. The breast was “evolving” similarly to the other breast, while patients’ body and weight were changing. This data has also been reported in the literature.

We have added this observation in the discussion:

“A significant correlation between change of weight and fat transplant volume survival over the years has been reported in the literature” line 255

and added a new reference:

Ueberreiter CS, Ueberreiter K, Mohrmann C, Herm J, Herold C. Langzeitevaluation nach autologer Fetttransplantation zur Brustvergrößerung [Long-term evaluation after autologous fat transplantation for breast augmentation]. Handchir Mikrochir Plast Chir. 2021 Apr;53(2):149-158. German. doi: 10.1055/a-1183-4338. Epub 2020 Aug 10. PMID: 32777824

Reviewer 2 Report

The manuscript is original because it explores the field of exclusive lipofilling in breast reconstruction after total mastectomy and evaluates the satisfaction and quality of life of the patients. It is written appropriately and is correctly designed, although the population examined is numerically low. 

I have only a suggestion for authors. You should eliminate the part regarding the literature review (text and tables) because it represents another kind of article and doesn't add anything to your argumentation.

Author Response

We thank the reviewer for his comments.

The number of patients is indeed low, however we thought it was important to enrich existing literature as current studies on this topic include a small number of patients as well. 

Concerning littérature review: we have added this part in our manuscript only after discussing with the editor. We think that it enables summarizing current data of already published studies as well as highlighting their major points.

However, it is likely to be removed if the editor shares the same opinion.

Reviewer 3 Report

The paper is interesting, the technique is good, and the results are good, but have some flaws.

The results seem nice, but figure 4 seems unclear to me as the reconstructed breast seems to be operated vertically (?, there is a scar). 

Why do the authors include a literature review if they have an own study?

It would be interesting to evaluate the reconstructed breasts by ultrasound regarding the number of cysts/fat necrosis. How is the cancer follow up done in these breasts, MRI, ultrasound, mammography? They should show pictures. Why were these patients not offered a "regular" reconstruction.

Explain why 10% had more than 3 operations, how the "immediate" reconstructions were done. Was the healing in smokers impaired? How did the breast volume react over the time? was there a difference in fat healing with regard to the harvest area? What about the stem cells? The authors cite Sorrentino's study, but what is the conclusion for their own patients? In Discussion/Conclusion the authors write: "radical mastectomy". Please explain why a radical mastectomy was performed (very unusual nowadays) or correct. 

In experienced centers a secondary autologous flap is done in one operation in 3-4h. So, autologous flap surgery is not the enemy here. It can be done immediately, so there is no loss of breast for the patient.  

I think that lipofilling has a big place as reconstructive technique but it needs better papers to promote it.

There are some papers that are not cited but seem to be interesting: 

Ueberreiter CS, Ueberreiter K, Mohrmann C, Herm J, Herold C. Langzeitevaluation nach autologer Fetttransplantation zur Brustvergrößerung [Long-term evaluation after autologous fat transplantation for breast augmentation]. Handchir Mikrochir Plast Chir. 2021 Apr;53(2):149-158. German. doi: 10.1055/a-1183-4338. Epub 2020 Aug 10. PMID: 32777824.

Chirappapha P, Rietjens M, De Lorenzi F, Andrea M, Hamza A, Petit JY, Garusi C, Martella S, Barbieri B, Gottardi A. Evaluation of Lipofilling Safety in Elderly Patients with Breast Cancer. Plast Reconstr Surg Glob Open. 2015 Aug 10;3(7):e441. doi: 10.1097/GOX.0000000000000411. PMID: 26301130; PMCID: PMC4527615.

Round 2

Reviewer 3 Report

Some of the suggestions were incorporated while others were just answered in the authors reply. I do not see the changes to the legend of fig 3. 

Author Response

We provide a point-by-point response to the reviewer’s comments : 

Comments 1: "Some of the suggestions were incorporated while others were just answered in the authors reply".

Reply: this time we have integrated the majority of the comments in the text.

Comment 2: I do not see the changes to the legend of fig 3.

Reply: the change had indeed been made only in the text. We have now added it in the figure legend.

All changes are highlighted in the new version of the text.

Please see the attachment (point-by-point response. + new manuscript)

Round 3

Reviewer 3 Report

I checked the paper and find it now suitable for publication.